# "Addressing the bigger picture": A qualitative study of internal medicine patients' perspectives on social needs data collection and use

Victoria H. Davis[1], Katie N. Dainty[2,3], Irfan A. Dhalla[2,4,5], Kathleen A. Sheehan[2,6,7], Brian M. Wong[5,8,9], Andrew D. Pinto[1,2,10,11] *

**1** Upstream Lab, MAP Centre for Urban Health Solutions, Li Ka Shing Knowledge Institute, St. Michael's Hospital, Toronto, Ontario, Canada, **2** Institute of Health Policy, Management and Evaluation, University of Toronto, Toronto, Ontario, Canada, **3** Department of Research and Innovation, North York General Hospital, Toronto, Ontario, Canada, **4** Department of Medicine, St. Michael's Hospital, Toronto, Ontario, Canada, **5** Department of Medicine, University of Toronto, Toronto, Ontario, Canada, **6** Department of Psychiatry, University of Toronto, Toronto, Ontario, Canada, **7** Centre for Mental Health, University Health Network, Toronto, Ontario, Canada, **8** Sunnybrook Health Sciences Centre, Toronto, Ontario, Canada, **9** Centre for Quality Improvement and Patient Safety, University of Toronto, Toronto, Ontario, Canada, **10** Department of Family and Community Medicine, St. Michael's Hospital, Toronto, Ontario, Canada, **11** Department of Family & Community Medicine, University of Toronto, Toronto, Ontario, Canada

* Andrew.pinto@utoronto.ca

**Data Availability Statement:** The data is only available upon request given institutional privacy and ethical restrictions on sharing it publicly.

## Abstract

### Background

There is increasing interest in collecting sociodemographic and social needs data in hospital settings to inform patient care and health equity. However, few studies have examined inpatients' views on this data collection and what should be done to address social needs. This study describes internal medicine inpatients' perspectives on the collection and use of sociodemographic and social needs information.

### Methods

A qualitative interpretive description methodology was used. Semi-structured interviews were conducted with 18 patients admitted to a large academic hospital in Toronto, Canada. Participants were recruited using maximum variation sampling for diverse genders, races, and those with and without social needs. Interviews were coded using a predominantly inductive approach and a thematic analysis was conducted.

### Results

Patients expressed that sociodemographic and social needs data collection is important to offer actionable solutions to address their needs. Patients described a gap between their ideal care which would attend to social needs, versus the reality that hospital-based teams are faced with competing priorities and pressures that make it unfeasible to provide such care. They also believed that this data collection could facilitate more holistic, integrated

According to our institutional privacy policy and research ethics, we are not able to publicly release our qualitative data as it contains potentially identifiable and confidential patient information. Reasonable data access requests can be considered by contacting Unity Health Toronto Research Ethics Board (researchethics@unityhealth.to).

**Funding:** The Canadian Institutes of Health Research (CIHR) SPOR PIHCI grant, held by Dr. Andrew Pinto, provided funding for participant honorariums only. The funders had no role in study design, data collection and analysis, decision to publish, or preparation of the manuscript.

**Competing interests:** The authors have declared that no competing interests exist.

care. Patients conveyed a need to have a trusting and transparent relationship with their provider to alleviate concerns surrounding bias, discrimination, and confidentiality. Lastly, they indicated that sociodemographic and social needs data could be useful to inform care, support research to inspire social change, and assist them with navigating community resources or creating in-hospital programs to address unmet social needs.

## Conclusions

While the collection of sociodemographic and social needs information in hospital settings is generally acceptable, there were varied views on whether hospital staff should intervene, as their priority is medical care. The results can inform the implementation of social data collection and interventions in hospital settings.

## Introduction

There have been increased efforts to expand the role of health systems to integrate social care into medical practice. This has been of interest due to recognition of the impact of sociodemographic characteristics (e.g., race, sexual orientation) and social needs (e.g., social support, food insecurity, housing) on health and healthcare [1]. For example, people with unmet social needs have poorer quality of life [2, 3], increased morbidity and premature mortality [4], and higher utilization of hospital resources [5–7].

To better identify patients' social factors, a greater number of healthcare settings are implementing systematic sociodemographic and social needs data collection across their entire patient population [8]. The systematic collection of sociodemographic and social needs information differs from the existing medical practice of taking the social history, which varies in quality and comprehensiveness, and is often not fully documented [9, 10]. Further, there are multiple opportunities to utilize this data beyond immediate patient care, which is the purpose of taking a social history.

Social data collection could facilitate more appropriate treatment and interventions to address patients' needs through community resource referrals [11, 12]. For example, patients who have been provided with assistance for their unmet social needs reported higher quality care and satisfaction [13], as well as patient knowledge of and access to supports [11]. The systematic collection of sociodemographic and social needs data could also inform the development of targeted quality improvement initiatives [14], health system changes and inform population health [15, 16].

Despite the potential benefits, there is a considerable knowledge gap to understanding patients' views on how sociodemographic and social needs data should be collected and used, particularly in inpatient internal medicine settings [8, 10, 17]. Inpatient internal medicine largely serves older adults with multiple complex chronic conditions who require a generalist approach to care [18, 19]. Most research concentrates on addressing social needs in primary care [7, 20, 21]; however, outpatient settings have different patient populations, staff workflow, resources, and opportunities to develop trusting relationships compared to inpatient settings [20]. In contrast, internal medicine may be the first point of contact for inpatients without primary care providers; many of these patients may experience greater barriers because of their social situations, and consequently rely on hospitals for treatment [22, 23]. Additionally, internal medicine inpatients are sicker and may have greater medical complications due to their

social needs [23]. Given hospital providers' expertise and experience caring for patients with unmet social needs, hospital providers are well-suited to understand the interaction between patients' needs and their acute medical illness.

To better understand opportunities to respectfully integrate social care into inpatient settings, a qualitative study was undertaken to ascertain patients' perspectives on the systematic collection and use of sociodemographic and social needs data in internal medicine.

## Methods

### Study design and setting

This research adopted an interpretive description methodology developed by Sally Thorne and colleagues to generate practical knowledge that can be applied to clinical practice [24, 25]. The Health Equity Implementation Framework (HEIF), specifically the domains of characteristics of the innovation and patient factors [26], partly guided the development of the interview questions. The HEIF prioritizes the societal influences that affect equitable implementation of innovations (e.g., social care integration) in healthcare settings and health disparities [26].

This study occurred at a Toronto hospital where approximately 40% of the neighborhood's patient population are non-white and 24% are low-income after tax [27, 28]. The patient population made this hospital an opportune location to explore diverse, heterogeneous perspectives on this topic. The COnsolidated criteria for Reporting Qualitative research (COREQ) guided the reporting of the study (S1 Table) [29]. Research ethics board approval was obtained at Unity Health Toronto (#21–259).

### Participants and recruitment

Eligible participants were admitted to inpatient general internal medicine, able to speak and understand English, capable of providing consent, and 18 years of age or older. Maximum variation sampling helped to capture heterogeneous cases and facilitate an understanding of common themes across patient characteristics [30], specifically, patients with or without unmet social needs, and diverse races and genders. This approach was selected to provide a range of perspectives, given that many healthcare institutions are starting to systematically collect sociodemographic and social needs data across the *entire* patient population (as opposed to only patients with "visible" unmet social needs). Thus, the perspectives of both patients with and without unmet social needs are important to inform future implementation.

Participants were approached face-to-face in the ward with the assistance of an attending internist. Participants had no pre-existing relationship with the interviewer (VHD), a female graduate student, who introduced participants to the study and their reasons for conducting the study. At the time of data collection, the interviewer had a B.Sc. and was a master's student. She had previous experience conducting qualitative health research and received extensive input from experts specifically related to the research methods and cultural sensitivity training.

### Data collection

To inform the sampling approach and to help participants develop experience answering sociodemographic and social needs questions, participants completed a sociodemographic and social needs questionnaire (S1 File) [31, 32]. The Hunger Vital Sign™ [33] was added. An open-ended interview guide was developed based on literature searches, the HEIF, and discussions with the research team (S2 File). The guide was refined prior to the interviews and after the first three interviews. Topics included: relevance of social needs on health and care, lifestyle and living circumstances, challenges after discharge, feelings being asked about

sociodemographic characteristics and social needs, the hospital's role in assisting patients with their social circumstances, and what should be done to address social needs. Following the interview, participants were offered a list of local community resources (e.g., shelter and food bank information, government assistance programs, financial assistance, legal aid, mental health supports). The interviews took place from December 2021 to February 2022.

Interviews were conducted at the patient's bedside. Some patients had private rooms due to their illness, and other interviews occurred in shared rooms where one or more non-participants were present. Interviews were audio recorded and detailed field notes were taken. Interviews were transcribed by Otter.AI, and the first author conducted an in-depth check for accuracy [34].

### Data analysis and management

Data analysis occurred in conjunction with data collection. The sample size was determined by thematic and participant saturation [35, 36]. For thematic saturation, a saturation table was used, with the main themes and subthemes listed in the row of the table and interviews were listed in the columns [35, 36]. Each interview documented whether new information arose and, if it did, an annotated summary of the new findings were described within that cell [36]. For participant saturation, a recruitment matrix was created to keep track of the maximum variation sampling, with the aim of at least one participant in each matrix cell.

Three transcripts were coded by VHD, after which the principal investigator (ADP) thoroughly reviewed the three transcripts and codes and provided detailed feedback. No major changes were made to the codes based on the review, and the codebook was agreed upon during discussions with the research team. A thematic analysis was conducted using a predominantly inductive approach [30, 34, 37], with NVivo software. The analysis consisted of examining, organizing, and coding segments of the interviews. From there codes were grouped into initial themes to address the study objectives, as described by Braun and Clarke [34]. These themes were reviewed in relation to the codes and data, and were defined and named [34]. The study team met regularly to discuss themes and findings to assist with the analytic process.

Throughout the entire research process, the interviewer regularly updated a reflexive diary to record personal introspections, expectations, and biases related to the research [38]. For example, she considered how her position as a white female of Irish descent and how her own financial and social privilege may influence the development of trust and communication about sensitive topics with diverse patients, as well as her personal beliefs about the importance of integrating social care in healthcare settings. Memos were also created during and after each interview to mitigate bias, as well as an audit trail [39].

## Results

### Participants

Eighteen interviews were conducted for participant saturation, and thematic saturation was reached at thirteen interviews. A total of 25 patients were deemed eligible for the study, however, some were unavailable at the time of recruitment (e.g., were undergoing diagnostic testing, preparing to be discharged). The participant sample was aligned with the older population in inpatient general internal medicine [19]. See Table 1, which was determined by self-report through the sociodemographic and social needs questionnaire. Sixty-six percent of participants did not have unmet social needs at the time of the interview (n = 10). Interviews were approximately 60 minutes long.

**Table 1. Qualitative interview participant characteristics (n = 18).**

| Characteristics | Patients (n = 18) |
|---|---|
| Age, mean years (standard deviation) | 66 (±11) |
| Female, *n* (%) | 7 (40%) |
| Race | |
| Black | 3 (17%) |
| East/Southeast Asian | 1 (6%) |
| South Asian | 1 (6%) |
| White/European | 12 (67%) |
| Multiple races | 1 (6%) |
| Highest education level | |
| ≤ High school diploma | 4 (22%) |
| Trades certificate/diploma | 4 (22%) |
| Postgraduate degree | 5 (28%) |
| College/university or some college/university | 5 (28%) |
| Presence of social needs at the time of the interview | |
| No | 10 (55.6%) |
| Yes | 8 (44.4%) |
| Key unmet social needs reported (of N = 8) | |
| Social, subsidized, rent-geared-to-income housing, or shelter | 7 (87.5%) |
| Difficulty making ends meet in last year | 6 (75%) |
| Some degree of food insecurity or worries about insecurity in last year | 4 (50%) |
| Missed utility bill payments in last year | 3 (37.5%) |
| Lack social support | 3 (37.5%) |
| Missed rent payments in last year | 2 (25%) |
| Missed an important appointment due to cost of transportation in last year | 2 (25%) |
| Employed | |
| No[a] | 13 (72.2%) |
| Yes | 5 (27.8%) |

[a]Most were retired (*n* = 10, 55.6%)

## Themes

Six main themes were developed to represent the data (Table 2).

**1. Feasible versus ideal: Gap between ideal care and perceptions about the care that hospitals can feasibly provide.** There was a gap between participants' perceptions about their ideal care and the care they perceive to be realistic and feasible for hospitals to provide. While participants found sociodemographic and social needs data collection to be acceptable and valuable to address unmet social needs, their views on the role of the hospital in acting on these needs varied. In other words, there was a clear juxtaposition and tension between what the hospital *should* do (i.e., collecting sociodemographic and social needs data and addressing unmet social needs), versus what the hospital *could* or *would* do. Participants were uncertain about whether the hospital has a role in collecting this information and addressing social needs due to implementation barriers (funding, political buy-in, available staff and time), and were cognizant that these implementation barriers could prevent providers from meaningfully addressing their needs.

**Table 2. List of themes and subthemes (n = 18).**

| Themes | Subthemes |
|---|---|
| 1. Feasible versus ideal: gap between ideal care and perceptions about the care that hospitals can feasibly provide | Role of the hospital versus government or community |
| 2. Only ask if you can act: Data collection should accompany actionable interventions | |
| 3. Understanding the bigger picture: Desire for holistic, integrated care that targets social needs | Lack of whole person care |
| | Continuity of care and care coordination |
| 4. Provider prejudice: Concerns about sharing personal information and its impact on care | Fear of health providers' perception or judgement |
| | Data collection can be invasive |
| | Varied perspectives on privacy and confidentiality |
| | Data could fuel provider bias and discrimination |
| 5. Trust before disclosure: Verbal data collection facilitates relationship-building and alleviates concerns | Communication and transparency facilitate trust |
| 6. More assistance, more contact numbers: Thoughts on how data can improve care and foster social change | Directly improve care |
| | Inform future hospital programs |
| | Increase access to and knowledge of community resources |
| | Contribute to societal understanding of inequities |

"Yeah, if you assume that hospital staff has the resourcing to do so. Like right now, they don't have a chance. There's no freaking way. But if they had the chance, yeah! It's all part of your general overall health and welfare. It would all tie together. . .So yeah, it would be beneficial that they all knew. And they could form a different viewpoint [on patient's care, and] do something about it. . .But you got to have the resources."

(P3)

*Role of the hospital versus government or community.* Similarly, some participants described a distinction between the role of the hospital and government or community, which influenced their perspectives about implementing interventions in inpatient internal medicine. For example, one participant was asked for their thoughts about the hospital staff connecting patients to community organizations, and indicated that the hospital, compared to the government, does not have time to intervene on social needs or community outreach. This connects to participants' views that the hospital may not be the most feasible setting for data collection and intervention.

"There's a need for that, for sure. You see that [during] the summertime with tents all over the parks and things, you know, [with] people living in them. . .Yeah, there's a definite need for help in that respect. I don't think hospitals would necessarily be involved in that. It is a government thing. . .I think [hospitals] are probably up to their neck now with what they have to do."

(P5)

"So I'm against it. You can only stretch so thin. You're doing one job. And now you're jumping into another one. Whereas you better go and get your community to jump in together and get your community to fund it. And that's the job for them."

(P4)

**2. Only ask if you can act: Data collection should accompany actionable interventions.** Participants supported data collection, so long as the hospital could provide meaningful resources and referrals to address their needs or improve their care. There was an emphasis on actionable interventions for social needs compared to sociodemographic data collection.

Some participants qualified the importance of data collection based on whether someone needed or wanted help for their social needs: "[It] depends on the patient's needs and depends on what a patient is wishing to divulge." (P10) However, many of these participants were uncertain as to how the hospital determines which patients to approach about their needs. Some mentioned that patients who are truly in need will ask the hospital for help, and that this was a feasible method to ensure that resources only go to those in greatest need.

> "I suppose in certain cases. . .but I don't think for the average person. . .You have to go to hospital to get healthcare."
>
> (P5)

**3. Understanding the bigger picture: Desire for holistic, integrated care that targets social needs.** Many participants expressed how their health is influenced by the social determinants, such as food security, income, housing, and access to healthcare. They wanted the hospital to provide them with holistic care and believed that the collection of sociodemographic and social needs information could facilitate this, by helping their providers know more about them as a person. Participants also made connections between the hospital only addressing symptoms of their condition, or biomedical causes, rather than trying to address causes that are tied to their social needs.

> "Knowing more background circumstances hopefully will let you address the bigger picture. . .In a lot of cases, all they're working with is what they can see now. Not see what led up to that situation."
>
> (P6)

> "To me, that's what healthcare is, it's holistic. . . [The hospital has] to know the whole of me, the whole of every patient."
>
> (P13)

*Lack of whole person care.* The concept of holistic care was also raised in the context of discussion about treating and caring for patients as an entire person. Some participants felt that the entire focus in the hospital was on providing "band-aid solutions" (P8) for their most immediate medical issue. Multiple participants felt that they were just "a number now, not [a] person" (P8).

> "It's part of life. It's the same as the hospitals, any hospitals, some of them here are really nice. They [provide care] as quick as they can to get [you] out of here. You're just a cog in a wheel. You're nobody. It's like everywhere."
>
> (P3)

> "And I know everybody is stressed out, everybody's like, 'oh, well, I've got more patients.' They don't have time for you anymore. . . [They] have to find a different way of how [they] interact with [their] patients and how [they] come across to [their] patients to make them

feel, 'hey, we're here. We care about you. Emotionally, we care about what's happening with you mentally.' I think [they should] say to the patient, 'what can I do to help you?'"

(P8)

*Continuity of care and care coordination*. Some participants identified the importance of continuity of care and care coordination: "It's that holistic care that is so important in today's world. I think we have to get rid of the segregation." (P13) Participants also emphasized the need for strong communication and connection of one's medical and social situation between their family physician, hospital physician, social worker, and community organizations. This demonstrated the importance of having a family physician assist with continuity and coordination of care:

"It would be great if there was the family doctor, the medical nursing team, the social work team of that hospital, but unfortunately, many people don't have a family doctor. You know, so they're missing one of the steps."

(P13)

For patients without a primary care provider:

"It would have to be someone at the hospital that takes that responsibility. . . I just think that so many people go to walk-in clinics. So, there's no medical person coordinating them, you know?"

(P13)

That said, patients also recognized the complexities involved in coordinating resources outside of hospital walls. Even with coordinated, integrated care, there are barriers to accessing social needs services outside of the hospital due to system design, resources, and staffing constraints.

**4. Provider prejudice: Concerns about sharing personal information and its impact on care.** *Fear of health providers' perception or judgement*. Participants were fearful about how their provider would perceive them if they revealed their social needs or sociodemographic information. This was often connected to power dynamics: "I feel [physicians] are usually judgmental, so [they] lack empathy." (P7) Some participants described how feelings of pride and shame prevented their disclosure of sociodemographic and social needs information.

"They think that people. . .are going to perceive them as a different person. Or because of that wealthy doctor looking at me now, pushing me through the door. . .They're afraid of [the] perception of what this higher hierarchy person thinks about them. If I tell them that I haven't eaten, they're going to look at me like, 'oh, you haven't eaten?' And look down on you."

(P8)

"I think it's a cultural thing too. You feel embarrassed because you have to feel dependent on somebody to ask. . .I don't ask for help because I feel like if I asked for help, you know, what are they going to say to me?.. So, you're kind of hiding. . .When I say hiding, that's the thing you don't want to discuss that much. And that's why a lot of people in my culture,

too, they will suffer and nobody knows. Because they don't want to talk about those kinds of things. Sometimes even your illness, you don't want to discuss."

(P17)

"And in the case of my mother, my mother is very secretive. My mom would want people to perceive her as this is who I am. Other immigrants think that if they talk about their status, or if they talk about their financial needs and stuff, they'll be looked at differently, they'll be treated differently. Because this is the land of opportunity."

(P8)

*Data collection can be invasive.* Some participants indicated that the hospital was being invasive: "It's none of [the hospital's] business. . .Stick to what you know." (P4)

"[When asked about which questions made the participant feel uncomfortable]: My sexuality, my financial status, where I'm living, you know, whether it's a house or an apartment or shelter. . . And then they wouldn't be asking you these things anyways unless they [were] going to help you, but I mean, it's just that it's personal and [I] feel that, 'oh, is that necessary?'"

(P18)

*Varied perspectives on privacy and confidentiality.* Participants also expressed concerns about how this data would be shared with others. One participant was "scared [the data collection would] turn back to haunt you" (P17) if it is included in their health record. Several participants were concerned that hospital staff would gossip about patients.

"And you see somebody, you tell them the problem. Then you go around the corner and [hear them] say 'Oh I just see that patient blah blah blah.' No, you don't do that because sometimes the patient will be passing by you, you don't even know."

(P17)

"[The] info age means information is the currency. Which means it can be abused too. . . So, [you] have to be very, very careful about some of this stuff. The potential is there to benefit everybody, partially the potential is also there to screw everybody too."

(P3)

While some participants were concerned about privacy breaches, others had no concerns about privacy and confidentiality, indicating that they had "nothing to hide."

*Data could fuel provider bias and discrimination.* The most common concern was that data collection could fuel discrimination against patients on the basis of race, sexual orientations, and social needs, to negatively impact their care. As a result, some participants, particularly those who have experienced discrimination, preferred to be asked about their background by someone not directly involved in their medical care (e.g., social worker).

However, participants' discomfort and willingness to disclose sociodemographic and social needs information depended on whether they believed that their providers could use the information in a positive way to treat their condition or help with their social needs. These participants expressed that their concerns could be alleviated by understanding the purpose of data collection.

"Because I think there's biases. . .they want to know, you know, so much about you. I would be concerned about them being bias[ed] because I wouldn't think that that would have anything to do with the care that they need to give you. . .[however], they might need to know if you're going to need certain help and how they could plan for you to get the necessary daily help, you know, [and it would be ok to ask in those circumstances]. It all depends on how you look at it and why they will be asking you."

(P18)

Non-white participants also expressed fear about being racially targeted by providers who ask about their sociodemographic background and social needs or would seek details for why they were chosen to be approached for the study. Other participants explicitly expressed fears about being targeted.

"Are you asking me this because I'm Black? And then [doctors or providers are] afraid to ask [patients] because how would I perceive the question? Do you ask this to your white fellow people? You know, are you doing it because I'm a minority? Do you do this for all [of] your patients or just because you see me in a different light, you know?"

(P8)

"I think we should ask all [patients about their sociodemographic background and social needs] so that you don't think because I look a certain way, they're going to ask me."

(P18)

**5. Trust before disclosure: Verbal data collection facilitates relationship-building and alleviates concerns.** Participants were more willing to disclose personal information if they have a trusting relationship with their providers; it also depends on "how long you know your physician." (P17).

Participants spoke about who should approach them about their background and assist with their needs, with a preference for those within their circle of care who are well trained and knowledgeable, as opposed to administrative staff.

"If they trust you, they may give you a hint of what they're going through. There's some people that are very. . . who may be asking the questions, they can sometimes be very rude. . . So you have to trust the person too."

(P17)

Some participants indicated that social workers should have a greater role in the hospital and should meet with all admitted patients. Non-white participants and/or those with unmet social needs expressed a preference for social workers to collect their sensitive information. Some participants also did not want their physician or nurse to ask them questions, because they felt it may undermine their focus on their medical care.

"In some ways, it would be good. But then you're going to ask yourself, well, if they're dipping into everything out there, how are they looking after you medically? Are they cutting your service short to do other stuff? Like hospitals should be hospitals, that's what you've come for. To get looked after properly, now you're going to start digging into housing and like, how good are you as a medical team, if you have to do everything else? I personally

think that it's good to have the people around to get that part. But I want my doctor to be my doctor."

(P8)

To facilitate trust, patients expressed greatest interest and comfort in verbally disclosing their personal information and demonstrated the importance of time. Participants indicated that verbal interviews made them feel heard, held their provider accountable to help with their needs, and removed logistical barriers such as difficulty using their hands to write or read. A couple of participants mentioned that collecting data through a structured questionnaire may not be appropriate because "people just may not want to do it or [are] unable to do it. So yeah, I don't think that will work for everybody. You know, you [have to] take time and talk with them." (P3).

*Communication and transparency facilitate trust.* Transparency about why the data is being collected, from whom it is being collected, and how it will be used and stored, was seen as essential by participants. For example, participants wanted to be provided a logical reason about why the data is being collected prior to disclosure.

"I think then you need some sort of a kind of a. . . warm kind of statement. . . 'The following questions may be uncomfortable, and we want to assure you that answering them won't [negatively] affect your healthcare.' I think that you've got to give them that kind of reassurance. Before you ask those questions. So, they know [they] can answer those."

(P13)

**6. More assistance, more contact numbers: Thoughts on how data can improve care and foster social change.**   Overall, participants believed that social needs information could be used by the hospital to help improve their care and assist them with accessing support, whereas they identified that sociodemographic information could inform research about health and society and foster societal changes.

*Directly improve care.* Participants believed that this data collection could positively impact the treatment provided to patients to fit their specific context and needs. Some participants also emphasized the importance of being cared for by staff who reflect the same cultural and sociodemographic diversity of the admitted patients.

"Not everybody's equal so by looking at this—their social needs, maybe certain people need certain needs. I think I saw somebody in the hospital and this guy definitely could use some help. You can. . .set up something for him depending on who needs it. . . Different care. Depends, you know. Like the guy in my room, he [uses drugs]. He could use different care."

(P1)

They mentioned that hospitals could use sociodemographic data to inform their hiring practices and to better incorporate patients' culture to improve their treatment:

"Are you hiring a West Indian person because of [patients'] West Indian background? Maybe they need to know more about what's happening with this person. Maybe I'll com-

municate with a person who's Black, or who's white, or who's Indian. Like, I'm Indian—I have a language barrier. Do you have that person in place that can help me?"

(P8)

"I'm a transgender person, how am I going to talk to you when you know nothing about me? Or you're looking at me like I'm different from you, you know, and they need to put the stuff in place to reflect who you're serving, your demographic, who you are."

(P8)

Similarly, two participants described how this data collection could shed light on existing biases and discrimination against certain groups of patients and hold providers accountable for equitable care, regardless of race, sexual orientation, or social needs. For example, in situations where providers would otherwise provide less attention and time toward patients of diverse backgrounds and social needs, sociodemographic and social needs data could hold providers accountable (e.g., if data were evaluated at a later time and compared to care outcomes):

"It could put some pressure on the staff to take care of you, because everybody has biases, whether in [the] hospital or not."

(P9)

*Inform future hospital programs*. Most participants were focused on how sociodemographic and social needs data collection could directly improve their own care. Fewer participants spoke about how this data collection could help develop longer-term initiatives. For example, a couple of participants indicated that data collection could help identify areas that the hospital could focus on for educational purposes, such as educating patients on how to navigate the health and social system, or providing lectures and workshops on healthy eating, and avoiding alcohol. Some participants acknowledged that the hospital already assists with some of these supports (e.g., dietician). Another participant suggested that it be used for the creation of social programs within the hospital:

"Whatever they need, everything for programs in an area. Because it's hard for people to see where they are. Social, food programs, housing programs. . . Like come down, there's food here for you. It's hard to find where it is."

(P15)

*Increase access to and knowledge of community resources*. Participants expressed their desire to have more information about community resources and to be given help to access these resources. They also identified the importance of hospital outreach and having the hospital follow-up with them after discharge to see if they need help accessing these services.

One participant described how the hospital was the "access point" (P6) for patients who are less wealthy and who have more difficulty accessing medical care. From this perspective, participants indicated that the hospital could have an active role in booking appointments for patients with external organizations or agencies, and to help them "get that little push to go" [access needed community resources] (P1).

"I wouldn't expect the hospital to hold the patient's hand, 24/7. But. . . more guidance, more support. You know, more assistance, more contact numbers, and [instructions on]

how to do this and how to do that. So not quite hold your hand, but you know, hiring people to maybe be you know, outreach, that kind of thing when people need it."

(P9)

However, participants indicated that being given a pamphlet of community resources only provides a "short-term solution" (P13). They desired more sustainable, long-term assistance to resolve their social situation. Similarly, participants mentioned that being provided with contact information for community resources is insufficient. There are numerous barriers to accessing such resources after discharge:

"You know, they just give me a referral or so they. . .sometimes people just go home and put it down, because they don't understand where to start. How to begin. You know. . .Especially when you have to [access those resources on your own], because I don't have a computer. When they say go and this tab and this tab for this and that, you don't understand what they say. And you don't want to. . .. for them to feel like you're stupid or something. When I say that, [it refers to how you] go home and you're wondering what should I do? You know sometimes family members are very busy. They don't have time to come around to help you, you know, so I think they should have a program set up for people to better understand the system."

(P17)

*Contribute to societal understanding of inequities*. Participants also indicated that the collection of sociodemographic information in hospitals could contribute to societal understanding of inequities; one way this can occur is through research or widespread information of inequalities. For example, a white participant spoke about racism embedded within the healthcare system. They mentioned Joyce Echaquan, an Indigenous woman who was racially abused by a nurse on camera in Quebec, Canada, and later died because her providers incorrectly assumed she was experiencing a narcotics withdrawal instead of pulmonary edema [40, 41]. This stood out as an example to represent how knowledge of sociodemographic characteristics can be powerful to help raise awareness of inequities. Another participant drew an analogy to the collection of race-based data by the Toronto police:

"I'm in favor of knowing about people. And, you know, the analogy to the police statistics. And the police have gotten in trouble in recent years for not collecting [race-based data]. . . The reason they don't collect it, didn't collect it, was because there was tremendous opposition to their collecting it, you know, 20 years or 30 years ago. I think it's incredible. How do you know who's affected if you don't study it?"

(P14)

Participants also indicated how sociodemographic data collection can be used to inform research on health disparities and the future treatment and development of initiatives to improve health. For example, one participant referenced how Black people were disproportionately affected by COVID-19. Other participants emphasized how this data could lead to larger societal changes:

"I think that it's important if it leads to a fairer society, so I think that could be an outcome. So people could see it more clearly, regardless of race or sexual orientation."

(P9)

## Discussion

This study examined patients' perspectives on sociodemographic and social needs data collection and use in inpatient internal medicine. Although there was support for collecting this information to improve care and help address patients' social needs, patients do not necessarily expect hospitals to implement systematic sociodemographic and social needs data collection. Similarly, studies in primary care and emergency department settings have reported that patients value this data collection [42–44] and believe that social needs data collection should depend on available resources [45]. This highlights a need for hospitals to develop partnerships with community organizations and attempt to ensure resources to address social needs are in place prior to collecting this data.

### Trust and patient-centered care

The findings suggest a need for establishing trust and rapport with providers as a prerequisite to disclosing personal information. Across healthcare settings, patients have reported that sociodemographic and social needs data collection helped them feel more comfortable disclosing their personal information, so long as they felt valued as a person and were given time, kindness, and respect [42–44, 46]. Relationship-building is essential to alleviate the commonly reported concerns of stigma, discrimination, and lack of privacy [47]. Without trust, provider training, and compliance with privacy and confidentiality regulations, this data collection could cause psychological harm to patients and perpetuate health inequities [47].

Relatedly, some participants wanted dedicated, trained staff, such as social workers, to approach patients about their social needs. However, the literature reports mixed results regarding who should collect sociodemographic and social needs information [48]. It is likely infeasible for hospital staff to spend the same amount of time that the interviewer did with inpatients to establish rapport and trust; thus, long-standing patient-provider relationships in primary care settings may be a distinguishing factor and limitation to considering data collection measures in inpatient internal medicine settings.

The findings also differed from recent literature on data collection modalities. Participants believed that data should be collected verbally, in a patient-centered manner. This finding may be influenced by the acuity of patients in inpatient internal medicine and their older age. While the preference for verbal data collection has been reported [49, 50], other studies in primary care have found patients are more forthcoming when completing self-administered surveys [51, 52]. Self-administered electronic surveys are more efficient; they can be integrated into the electronic medical record and provide automated resource referrals [53–55]. Our research suggests that a "one-size-fits-all" approach to data collection is unlikely to be appropriate based on the diversity of patient populations [48], alongside considerations for possible trade-offs between the feasibility and acceptability of data collection modalities in inpatient settings.

### Racism and equitable implementation of data collection

Participants also provided suggestions for more equitable implementation of sociodemographic and social needs data collection, in response to fears about discrimination. It is imperative that data collection efforts confront the pervasive racism and discrimination in the healthcare system [56–59]. A commentary titled "Warning: we're integrating social care into a racially-biased health system", emphasized the need for deliberate efforts to prevent harm toward patients who are non-white and have intersectional identities; otherwise, there is a risk of perpetuating the inequities that health systems are intending to alleviate [56].

In this study, participants expressed concerns about being racially targeted by providers and may have an additional layer of mistrust when being approached about their sociodemographic and social needs information. As a result, some participants supported systematically collecting data from all patients, which aligns with research suggesting that this approach could also identify individuals with "hidden" unmet social needs [48, 56]. Further, some participants emphasized the importance of being cared for by staff who reflect the same cultural and sociodemographic diversity of the admitted patients. This likely also extends to the staff collecting this sensitive information. Additional, important considerations include: advocating for equitable system-level changes; staff sensitivity, ongoing skills training and curricula on racial biases and intersectionality; and meaningful partnerships and co-design of data collection and interventions, including a strong need for Indigenous data governance and sovereignty [56, 59–61].

## Meaningful solutions to address social needs

One of the most important findings was the gap between participants' perspectives of their ideal care and their perspectives on the care that is feasible for the hospital to provide. Participants identified that the hospital's sole focus on their medical needs left many important components of their health unaddressed, as a "band-aid solution" for their problems. Despite this, participants did not expect hospitals to integrate social care, due to time, staffing, and other constraints. These barriers are consistent with those observed across various healthcare settings [43, 62–64]. Integrating social care in healthcare settings remains a relatively new phenomenon for many patients and contributes to some patients questioning the hospital versus the government role in prevention and addressing unmet social needs.

This presents a duality: while hospitals may have a role in systematically collecting sociodemographic and social needs data and intervening on those needs, the lack of infrastructure and coordination between hospital and social services diminishes that role. Consequently, patients who lack primary care providers and have unmet social needs may "fall through the cracks", without the necessary support upon discharge from the hospital. Thus, the biomedical focus in hospital settings may be misaligned with the expressed needs and desires of patients to have a more holistic, integrated care experience.

Relatedly, the level of hospital support for addressing patients' unmet social needs matters to patients. For example, many participants did not expect the hospital to intervene on their social needs but had generally favourable views regarding the provision of community resource referrals. However, at the same time, participants were aware that community resource referrals are short-term solutions and insufficient to meaningfully address their needs. Some participants with unmet social needs experienced barriers to accessing support and desired follow-up assistance from hospital workers. Prior research has reported similar barriers, with estimates that just over one-third of individuals given a referral actually receive help [48, 65, 66]. Facilitators to helpful referrals include outreach attempts from care providers and persistent follow-up assistance [46, 67, 68] and providing tailored resources [46]. For example, a study in an outpatient pediatric clinic found that only 43% of families who received referrals were successful, and families that had four or more outreach sessions were 1.92 times more likely to have received a helpful referral [68]. Notably, this type of follow-up assistance after hospital discharge is suited for primary care settings, and strong coordination between hospital care and primary care is important to efficiently assist patients with their unmet social needs. However, this becomes more complicated with patients who lack primary care providers and experience multiple barriers to accessing one.

Even with outreach and follow-up, integrating social care with hospital services requires investment in government programs and social services in the community. For some participants who have unmet social needs, their experiences being denied social programs or community services left them skeptical that the hospital could change their social conditions, which serves as a deterrent to future attempts to accessing resources [69–72]. This has been referred to as a potential "road to nowhere problem" given the significant financial strain on social and community services in the US [72]. In addition to discrimination and shame, negative experiences accessing resources may contribute to the large gaps between patients' identified unmet social needs and their interest in receiving assistance in healthcare settings [69–71]. In this study, most patients indicated that they wanted more support and assistance but did not expect it. This emphasises the nuances and complexities involved with hospitals attempting to address patients' social needs, and questions remain about how to ensure the sustainability and availability of meaningful community resources and social programs [71, 73]. Despite these questions, a strong argument can be made that hospitals can work with partners (e.g., community organizations, schools and businesses) to try to provide services and resources that meet the social needs of patients. For example, hospitals can partner with charitable organizations to ensure that patients receive income supports that they are eligible for [74, 75] and work in partnership to increase the availability of affordable housing [75, 76].

## Strengths and limitations

This study helped to bridge important knowledge gaps related to the collection and use of sociodemographic and social needs information in hospital settings. Previous literature on this topic focuses on patients' perspectives on addressing social needs, particularly through community resource referrals (e.g., see 2,41,48). However, in this study, patients identified the value of such data collection to support health equity, even though it may not benefit themselves, and demonstrated empathy for others. Further, this research occurred within inpatient internal medicine, which is a novel setting for this inquiry with an underexplored patient population composed of older adults. Another strength is the maximum variation sampling approach, which facilitated diverse patient perspectives across a range of sociodemographic backgrounds and patients with and without unmet social needs. The main limitations are that non-English speakers were excluded, and that it occurred at one hospital location, which may limit the applicability of our findings to other demographics and regions.

Future research should extend this study to include non-English speakers and additional groups, including Indigenous people. It should also be expanded to other areas, hospital sizes, and types, to understand patients' perspectives across contexts. Lastly, it is important to explore the perspectives of interdisciplinary internal medicine providers on data collection to inform implementation efforts.

## Conclusions

Participants expressed that sociodemographic and social needs data collection could be used to improve patient care, inform the provision of social needs resources and societal understanding of inequities. However, their views varied on the hospital's role in data collection and assisting with social needs, because of infrastructure, time, staffing, and funding barriers. This reflected the lack of holistic care that incorporates their sociodemographic and social needs information, which some participants wanted as part of their hospital experience. These results can inform the implementation of social needs integration in hospital settings.

## Supporting information

**S1 Table. Consolidated criteria for reporting qualitative studies (COREQ) checklist.**
(DOCX)

**S1 File. Sociodemographic and social needs questionnaire.**
(DOCX)

**S2 File. Interview guide.**
(DOCX)

## Acknowledgments

We would like to sincerely thank the hospital patients who generously contributed their time to participate in this research, and for their openness in sharing their personal stories and perspectives. We are also grateful for the time and assistance of many internists who assisted with our recruitment in the ward.

## Author Contributions

**Conceptualization:** Victoria H. Davis, Katie N. Dainty, Irfan A. Dhalla, Kathleen A. Sheehan, Brian M. Wong, Andrew D. Pinto.

**Formal analysis:** Victoria H. Davis, Katie N. Dainty, Andrew D. Pinto.

**Funding acquisition:** Andrew D. Pinto.

**Investigation:** Victoria H. Davis.

**Methodology:** Victoria H. Davis, Katie N. Dainty, Kathleen A. Sheehan, Andrew D. Pinto.

**Project administration:** Victoria H. Davis.

**Resources:** Irfan A. Dhalla, Andrew D. Pinto.

**Software:** Victoria H. Davis.

**Supervision:** Andrew D. Pinto.

**Validation:** Victoria H. Davis, Katie N. Dainty, Andrew D. Pinto.

**Writing – original draft:** Victoria H. Davis.

**Writing – review & editing:** Victoria H. Davis, Katie N. Dainty, Irfan A. Dhalla, Kathleen A. Sheehan, Brian M. Wong, Andrew D. Pinto.

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
