## [Decision Letter · Decision Letter 0]

10 Apr 2023

PONE-D-23-05027"Addressing the bigger picture": qualitative study of internal medicine patients' perspectives on social needs data collection and usePLOS ONE

Dear Dr. Pinto,

Thank you for submitting your manuscript to PLOS ONE. After careful consideration, we feel that it has merit but does not fully meet PLOS ONE’s publication criteria as it currently stands. Therefore, we invite you to submit a revised version of the manuscript that addresses the points raised during the review process.

We look forward to receiving your revised manuscript.

Kind regards,

André Ramalho, PhD

Academic Editor

PLOS ONE

Journal Requirements:

2. We noted in your submission details that a portion of your manuscript may have been presented or published elsewhere. [A version of this manuscript is part of the lead author's master's thesis, which is under embargo and will be publicly released in 2024. ] Please clarify whether this [conference proceeding or publication] was peer-reviewed and formally published. If this work was previously peer-reviewed and published, in the cover letter please provide the reason that this work does not constitute dual publication and should be included in the current manuscript.

Additional Editor Comments:

Dear Authors,

I was pleasantly surprised by the approach demonstrated in your article. Congratulations on taking the initiative to work on concepts that are often overlooked in the hospital environment. Your conclusion highlights the critical importance of addressing social needs and collecting sociodemographic data to improve patient care and provide resources to address their social needs. As we know, the natural history of disease and health is influenced not only by biological factors but also by social and environmental factors.

Your article also brought to light the challenges hospitals face in fulfilling this role, such as infrastructure, time, staffing, and financial barriers. Furthermore, the article emphasizes the need for an integrated approach that incorporates sociodemographic information and patients' social needs to provide adequate care. Therefore, it is vital for hospitals to adopt a broader view of health that recognizes their responsibility is not limited to diagnosing and treating diseases, but also includes understanding the social and environmental conditions that impact patients' health.

Incorporating this understanding must be done in conjunction with other levels of care in the network, including primary health care, which plays a crucial role in identifying and addressing the social and environmental needs of patients. I suggest considering these points in your discussion.

Hospitals can work outside their walls in various ways to address the social and environmental factors that impact patients' health. Here are a few possibilities:

1. Partnerships with other institutions: By partnering with community organizations, schools, and businesses, hospitals can provide services and resources that meet the social and environmental needs of patients.

2. Awareness programs: Hospitals can offer awareness programs on social and environmental factors that impact patients' health, such as nutrition, physical activity, and disease prevention.

3. Support programs: Hospitals can offer support programs to help patients address social and environmental problems, such as unemployment, inadequate housing, and domestic violence.

4. Integration with other levels of care network: Hospitals can integrate with other levels of care in the network, such as primary health care, to identify and address patients' social and environmental needs more efficiently and effectively.

These are just a few of the possibilities for hospitals to act outside their walls and consider the social and environmental factors that impact patients' health. It is crucial for hospitals to consider the specific needs of their community and work collaboratively with other institutions to provide adequate and effective services and resources.

Thank you for the opportunity to share my thoughts, and I hope this feedback is helpful.

Best Regards,

André Ramalho, MD PhD

Academic Editor

PLOS One

Reviewers' comments:

Reviewer's Responses to Questions

**Comments to the Author**

1. Is the manuscript technically sound, and do the data support the conclusions?

Reviewer #1: Yes

Reviewer #2: Yes

2. Has the statistical analysis been performed appropriately and rigorously? 

Reviewer #1: N/A

Reviewer #2: Yes

3. Have the authors made all data underlying the findings in their manuscript fully available?

Reviewer #1: No

Reviewer #2: Yes

4. Is the manuscript presented in an intelligible fashion and written in standard English?

Reviewer #1: Yes

Reviewer #2: Yes

5. Review Comments to the Author

Reviewer #1: Itz a well investigated and well written exploration. However as a reader, I may suggest some minor additions to add more qualitative flavor.

Line 121-Kindly add a sentence about what criteria made a participant being ELIGIBLE.

Line 144-In the same way what do you mean by LOCAL COMMUNITY RESOURCES?

Line 131 & Line 150 describes interviewer. This description may be unified at the same place.

Line 185+ describes percentages where as in qualitative research, facts not figures matter.

Table-1 again focusses on figures and is too lengthy. It may be reduced to only important attributes of the participants.

IMPORTANT: A table specifying themes, subthemes and codes may kindly be included.

Reviewer #2: "Addressing the bigger picture": qualitative study of internal medicine patients' perspectives on social needs data collection and use

Title: Qualitative Study? Type: Semi structured interview.

Appropriate but long

Abstract: Appropriate and informative, but long. Method section did not mention the tools used in the study. No specific points were mentioned/described in the results section.

Keywords: Add more words like; University of Toronto.

Keywords are not included in the manuscript file after the abstract.

Introduction: Appropriate and informative; but long.

Objective: Appropriate and informative; SMART.

Methods: Appropriate and informative.

Results: Text of the Results; Appropriate and Informative. Well-structured.

Discussion: Appropriate and Informative. Section “Strengths and limitations” is well-organized and informative.

Conclusion: Appropriate and Informative.

References: Most of references are recent (Published <6 years ago).

Table 1: Appropriate and Informative.

6. PLOS authors have the option to publish the peer review history of their article (what does this mean?). If published, this will include your full peer review and any attached files.

Reviewer #1: No

Reviewer #2: **Yes: **Mohamed Farouk Allam

---

## [Author Response · Author response to Decision Letter 0]

18 Apr 2023

Dear Dr. André Ramalho and reviewers:

We are pleased to submit our revised article, “Addressing the bigger picture”: A qualitative study of internal medicine patients’ perspectives on social needs data collection and use”, for your consideration. We wish to thank you and the reviewers for your insightful comments and for allowing us to submit our revision. We are grateful for your time and constructive feedback.

To answer the comments on journal requirements, we have reviewed the PLOS ONE requirements and made changes accordingly. We have added additional references based on the reviewer comments. The research in the manuscript has not been formally published or peer-reviewed elsewhere. It was presented as a poster at a conference (Canadian Association for Health Services and Policy Research, 2022). We disclosed in our original cover letter that a variation of the manuscript was part of the lead author’s master’s thesis. It was a requirement for the completion of their degree that the thesis becomes publicly available and will be available in 2024 once the embargo ends. The manuscript submitted to PLOS ONE is not a dual publication. 

Lastly, regarding the comment on data availability, we have carefully considered PLOS ONE’s data access requirements and confirm that data is only available upon request given privacy and ethical restrictions on sharing it publicly. According to our institutional privacy policy and research ethics, we are not able to publicly release our qualitative data as it contains potentially identifiable and confidential patient information. Reasonable data access requests can be considered by contacting Unity Health Toronto Research Ethics Board (researchethics@unityhealth.to).

We have listed the comments from the editor and reviewers and responded below. 

Editor comments: 

Additional Editor Comments:

1. COMMENT: I was pleasantly surprised by the approach demonstrated in your article. Congratulations on taking the initiative to work on concepts that are often overlooked in the hospital environment. Your conclusion highlights the critical importance of addressing social needs and collecting sociodemographic data to improve patient care and provide resources to address their social needs. As we know, the natural history of disease and health is influenced not only by biological factors but also by social and environmental factors.

Your article also brought to light the challenges hospitals face in fulfilling this role, such as infrastructure, time, staffing, and financial barriers. Furthermore, the article emphasizes the need for an integrated approach that incorporates sociodemographic information and patients' social needs to provide adequate care. Therefore, it is vital for hospitals to adopt a broader view of health that recognizes their responsibility is not limited to diagnosing and treating diseases, but also includes understanding the social and environmental conditions that impact patients' health. 

Incorporating this understanding must be done in conjunction with other levels of care in the network, including primary health care, which plays a crucial role in identifying and addressing the social and environmental needs of patients. I suggest considering these points in your discussion.

Hospitals can work outside their walls in various ways to address the social and environmental factors that impact patients' health. Here are a few possibilities:

1. Partnerships with other institutions: By partnering with community organizations, schools, and businesses, hospitals can provide services and resources that meet the social and environmental needs of patients.

2. Awareness programs: Hospitals can offer awareness programs on social and environmental factors that impact patients' health, such as nutrition, physical activity, and disease prevention.

3. Support programs: Hospitals can offer support programs to help patients address social and environmental problems, such as unemployment, inadequate housing, and domestic violence.

4. Integration with other levels of care network: Hospitals can integrate with other levels of care in the network, such as primary health care, to identify and address patients' social and environmental needs more efficiently and effectively.

These are just a few of the possibilities for hospitals to act outside their walls and consider the social and environmental factors that impact patients' health. It is crucial for hospitals to consider the specific needs of their community and work collaboratively with other institutions to provide adequate and effective services and resources.

RESPONSE: Thank you very much for your thoughtful feedback and suggestions for our manuscript. We also appreciate that you share our interest on this important topic, and that you have provided additional points on what hospitals can do to address social needs in innovative ways. We have incorporated your feedback into the discussion section of the manuscript, including:

Page 27: “… and strong coordination between hospital and primary care is important to efficiently assist patients with their unmet social needs.”

Page 28: “Despite these questions, a strong argument can be made that hospitals can work with partners (e.g., community organizations, schools and businesses) to try to provide services and resources that meet the social needs of patients. For example, hospitals can partner with charitable organizations to ensure that patients receive income supports that they are eligible for [74,76] and work in partnership to increase the availability of affordable housing [75,76,78].”

Reviewers' comments:

REVIEWER #1: 

1. COMMENT: It’s a well investigated and well written exploration. However as a reader, I may suggest some minor additions to add more qualitative flavor. Line 121-Kindly add a sentence about what criteria made a participant being ELIGIBLE.

RESPONSE: Thank you for this comment. Original line 121 specifies what criteria makes a participant eligible, i.e., if they were “admitted to inpatient general internal medicine, able to speak and understand English, capable of providing consent, and 18 years of age or older”.

2. COMMENT: Line 144-In the same way what do you mean by LOCAL COMMUNITY RESOURCES?

RESPONSE: We have added clarification on page 29: “(e.g., shelter and food bank locations, government assistance programs, financial assistance, legal aid, mental health supports).”

3. COMMENT: Line 131 & Line 150 describes interviewer. This description may be unified at the same place.

RESPONSE: I have unified the descriptions of the interviewer to one place (under “participants and recruitment”). 

4. COMMENT: Line 185+ describes percentages where as in qualitative research, facts not figures matter. Table-1 again focusses on figures and is too lengthy. It may be reduced to only important attributes of the participants.

RESPONSE: We appreciate your focus on the qualitative components of the paper. We have removed the following sentence: “The most commonly reported unmet social needs were social, subsidized, or rent-geared-to-income housing or living in shelters (n=7) and difficulty making ends meet (n=6).” 

In response to your comment about Table 1, we have removed information reporting participant’s physical, emotional, or mental difficulties, and the type of difficulties. 

5. COMMENT: IMPORTANT: A table specifying themes, subthemes and codes may kindly be included.

RESPONSE: We have included Table 2, which has a list of themes and subthemes. Our style of qualitative research involves providing more detailed themes and subthemes as opposed to providing a more “general” topic/category for our themes. For example, theme five is: “Trust before disclosure: Verbal data collection facilitates relationship-building and alleviates concerns”. As a consequence, some of our codes are repetitive with our subthemes, and would not necessarily offer meaningful insight into the research. For example, the sub-theme “increase access to and knowledge of community resources” under theme 6 is a longer version of a couple of codes: community resources – knowledge, community resources – access, etc. Please let us know if this is something that you would still like us to add. 

REVIEWER #2: 

6. COMMENT: Title: Qualitative Study? Type: Semi structured interview. Appropriate but long

RESPONSE: We recognize the title is a bit long, but have made it that way to describe the content of the paper in sufficient detail while also reflecting participant voices – i.e., “Addressing the bigger picture”. (Please note that we have reread the title and have decided to add “A” in front of “qualitative study” for grammar.)

7. COMMENT: Abstract: Appropriate and informative, but long. Method section did not mention the tools used in the study. No specific points were mentioned/described in the results section.

RESPONSE: We have shortened the abstract by removing: “recorded with field notes and audiotapes, transcribed verbatim, and..” We did not add a description of the Health Equity Questionnaire because of word constraints. It was solely for the purpose of helping us understand the purposive recruitment sampling and to help patients understand what we meant by sociodemographic and social needs data collection. With regards to our results section in the abstract, we briefly described all of our six themes. 

8. COMMENT: Keywords: Add more words like; University of Toronto. Keywords are not included in the manuscript file after the abstract.

RESPONSE: Thank you - we have added more keywords and included it in the manuscript file after the abstract.

9. COMMENT: Introduction: Appropriate and informative; but long.

RESPONSE: Thank you for the feedback. We have carefully considered the introduction and opportunities to shorten it without removing necessary background from this field and addressing the significance and gaps of this research question. Please let us know if there is a particular part of the introduction that you feel should be shortened. 

10. COMMENT: Objective: Appropriate and informative; SMART. Methods: Appropriate and informative. Results: Text of the Results; Appropriate and Informative. Well-structured. Discussion: Appropriate and Informative. Section “Strengths and limitations” is well-organized and informative. Conclusion: Appropriate and Informative. References: Most of references are recent (Published <6 years ago). Table 1: Appropriate and Informative.

RESPONSE: Thank you very much for reviewing our paper and for providing feedback on each of our sections – we greatly appreciate your time.

---

## [Editor Report · Decision Letter 1]

2 May 2023

"Addressing the bigger picture": A qualitative study of internal medicine patients' perspectives on social needs data collection and use

PONE-D-23-05027R1

Dear Dr. Pinto,

I hope this message finds you well. I am delighted to let you know that your responses to the editor and reviewers' inquiries have been received and considered carefully.

So, we’re pleased to inform you that your manuscript has been judged scientifically suitable for publication and will be formally accepted for publication once it meets all outstanding technical requirements.

Kind regards,

André Ramalho, PhD

Academic Editor

PLOS ONE

---

## [Editor Report · Acceptance letter]

11 May 2023

PONE-D-23-05027R1 

“Addressing the bigger picture”: a qualitative study of internal medicine patients’ perspectives on social needs data collection and use 

Dear Dr. Pinto:

I'm pleased to inform you that your manuscript has been deemed suitable for publication in PLOS ONE. Congratulations! Your manuscript is now with our production department. 

Kind regards, 

on behalf of

Prof. Dr. André Ramalho 

Academic Editor

PLOS ONE